# A TOSCA-Based Conceptual Architecture to Support the Federation of Heterogeneous MSaaS Infrastructures †

Paolo Bocciarelli *,‡ and Andrea D'Ambrogio *,‡

Department of Enterprise Engineering, University of Rome Tor Vergata, Via del Politecnico 1, 00133 Rome, Italy
* Correspondence: paolo.bocciarelli@uniroma2.it (P.B.); dambro@uniroma2.it (A.D.)
† This paper is an extended version of our paper "ArTIC-M&S: An Architecture for TOSCA-based Inter-Cloud Modeling and Simulation" published in the Proceedings of 2020 Winter Simulation Conference, Orlando, FL, USA, 14–18 December 2020; pp. 2018–2029.
‡ These authors contributed equally to this work.

**Abstract:** Modeling and simulation (M&S) techniques are effectively used in many application domains to support various operational tasks ranging from system analyses to innovative training activities. Any (M&S) effort might strongly benefit from the adoption of service orientation and cloud computing to ease the development and provision of M&S applications. Such an emerging paradigm is commonly referred to as *M&S-as-a-Service (MSaaS)*. The need for orchestrating M&S services provided by different partners in a heterogeneous cloud infrastructure introduces new challenges. In this respect, the adoption of an effective *architectural approach* might significantly help the design and development of MSaaS infrastructure implementations that cooperate in a federated environment. In this context, this work introduces a *MSaaS reference architecture (RA)* that aims to investigate innovative approaches to ease the building of inter-cloud MSaaS applications. Moreover, this work presents *ArTIC-MS*, a conceptual architecture that refines the proposed RA for introducing the TOSCA (topology and orchestration specification for cloud applications) standard. ArTIC-MS's main objective is to enable effective portability and interoperability among M&S services provided by different partners in heterogeneous federations of cloud-based MSaaS infrastructure. To show the validity of the proposed architectural approach, the results of concrete experimentation are provided.

**Keywords:** MSaaS; M&S-as-a-service; reference architecture; FAIR; TOSCA



## 1. Introduction

Modeling and simulation (M&S) is a widely used approach to analyze systems, natural phenomena, and processes [1,2]. It encompasses the specification of a simulation model describing the addressed system or process from the required perspective, and the consequent implementation of the corresponding executable simulation system, which is run to reproduce the behavior of the system or process under study [3].

M&S approaches are neither tied to any specific application domain nor constrained to the adoption of a given development paradigm. Indeed, they have proven their effectiveness in various application domains. Among others, the development of *complex systems*, such as distributed systems, high-performance computing systems, or cyber–physical systems (CPS) might benefit from the adoption of a M&S approach as it allows analysts to evaluate different design alternatives before starting the actual system implementation. Specifically, innovative approaches, such as those that exploit methods and standards form the model-driven engineering (MDE) field, can be introduced to automate and ease the generation of a simulation model from the design models of the system under study [4–6].

Moreover, M&S approaches might be effectively adopted to support the development of *digital twins* (DTs) [7], which have recently gained great popularity among researchers and practitioners in the simulation field. A DT is a dynamic digital representation

(e.g., a simulation model) of a physical system, which is referred to as a *physical twin* (PT). The fundamental aspect of this paradigm is that the DT and the related PT are constantly aligned so that changes in the operational state of the PT are reflected in relevant changes in the DT configuration parameters. In this context, M&S approaches might be introduced to derive a DT from the design model of the addressed system, thus ensuring the initial compliance of the DT with its physical counterpart [8].

In order to leverage the potential benefits of M&S, technologies such as *cloud computing* and *service-oriented architectures (SOAs)* can be introduced to build and provide M&S applications through the composition of services available in the cloud, according to the emerging *M&S-as-a-service (MSaaS)* paradigm [9,10]. Thus, according to a MSaaS perspective, an M&S application can be built by integrating and orchestrating existing M&S services, to enhance the interoperability, composability, reusability, and cost-effectiveness of the M&S effort [11].

That is, the delivery of service composition in the cloud is a non-trivial task that requires the proper execution of the following activities:

1. Starting with the simulation requirements and objectives, a candidate *set of component M&S services* has to be *identified*, with each component service providing an interface that meets the composite service requirements;
2. A choreography-based or an orchestration-based *composition model* has to be specified to handle the execution flow of several component services;
3. Each component service has to be *configured and deployed* onto an execution platform, which, in turn, requires the integration of a set of hardware and software resources, such as computational nodes, containers, applications, networks connections, databases, and middleware.

As discussed in Section 4.1, in this work, we use the term *application orchestration* with reference to the aforementioned activities 1 and 2, while activity 3 is referred to as *infrastructure orchestration*.

The development of a MSaaS application becomes even more complex when the needed composition includes M&S services provided by different and heterogeneous cloud implementations in a so-called *MSaaS-federated infrastructure*.

In this context, the paper introduces an architecture for TOSCA-based inter-cloud modeling and simulation (ArTIC-MS), a conceptual architecture that aims to address the fair principles (findability, accessibility, interoperability, and reusability) in the development of a MSaaS platform.

Specifically, the contribution proposed in this paper, which extends our previous work [12], can be summarized as follows:

- The detailed description of the conceptual approach through which ArTIC-MS is specified;
- The complete specification of the ArTIC-MS functional view, by identifying the building blocks composing the conceptual architecture and introducing the relevant capabilities they provide;
- The specification of the ArTIC-MS operational view, which is outlined throughout the description of actors and relevant use cases.

Regarding the conceptual approach at the basis of ArTIC-MS specification, this paper introduces a *MSaaS reference architecture (RA)* that aims at identifying the abstract components and related capabilities required for addressing the development of MSaaS applications in a federated infrastructure. Then, the proposed RA derives the conceptual architecture. ArTIC-MS exploits TOSCA (topology and orchestration specification for cloud applications) [13] as an enabling standard to effectively ease the development of M&S applications by integrating simulation components provided by different partners, and deployed onto heterogeneous cloud infrastructure, in order to maximize the reuse of existing components.

Regarding the functional specification of ArTIC-MS, it should be underlined that, from a general perspective, its capabilities allow simulation developers to cope with

both the infrastructure and application orchestration. The latter has been investigated in previous contributions, where approaches, tools, and standards in the MDE field have been introduced to support the generation of executable simulations from abstract orchestrations, such as [14]. Thus, this aspect is not further discussed in this work, which focuses on infrastructure orchestration.

From a concrete point of view, this work also discusses an example application where a MSaaS prototype implementation based on ArTIC-MS was developed to assess how the proposed approach might effectively contribute to satisfying the FAIR principles of a MSaaS application in a federation of heterogeneous cloud infrastructure. The experimentation was carried out within the scope of the *MSaaS Architectures and Services for training and experimentation (MASTER)* project, a research effort carried out under the National Program for Defense Research. The MASTER project aims at identifying innovative solutions for developing M&S applications in the cloud. Specifically, the addressed case focuses on the investigation of approaches to promote the interoperability of M&S resources. In this respect, the proposed example deals with the integration of the ArTIC-MS prototype with the open cloud ecosystem application (OCEAN) [15], a MSaaS platform based on the open-source OpenStack cloud infrastructure [16].

The rest of this paper is structured as follows. Section 2 revises the existing literature and clarifies the novelty of the proposed contribution, Section 3 provides the concepts at the basis of this work by briefly outlining the TOSCA standard and the OpenStack IaaS cloud implementation, Section 4 outlines the adopted architectural approach and introduces the MSaaS RA, Section 5 illustrates the ArTIC-MS conceptual architecture, Section 6 provides an operational description of ArTIC-MS throughout the specification of relevant use cases, Section 7 discusses the experimentation. Finally, Section 8 provides concluding remarks.

## 2. Related Work

Various contributions can be found that demonstrate how simulation-based techniques have been successfully adopted for years in many application domains to support different operational needs [17–21]. As an example, the roles of M&S technologies and the MSaaS paradigm as key enablers to develop innovative training capabilities, support system analysis, and decision-making were underlined in [22].

Regarding the investigation of abstract architectures which specifically aim at addressing the development of MSaaS infrastructure, relevant contributions can be found in [10,12,23].

In [10,23] the outcomes of the NATO Modeling and Simulation Groups MSG-136 have been presented. Such contributions identify the preliminary requirements of a possible MSaaS reference architecture for supporting the NATO operational needs, specifically with regard to interoperability issues. In this respect, this paper shares the common objective of identifying a reference architecture for MSaaS applications. On the other hand, this work goes beyond: along with the description of a conceptual framework, the TOSCA standard is also introduced for ensuring interoperability and supporting the development of MSaaS applications in federated cloud infrastructure. Moreover, unlike the above-mentioned contributions, ArTIC-MS is not tied to any application domain and can be effectively adopted in various operational contexts.

Regarding the development of MSaaS platforms, several contributions have been proposed that address such an issue from different perspectives [17,24–26]. In [17,24], interoperability and composability have been identified as two of the most relevant challenges for M&S systems. While interoperability is defined as the ability to exchange data among simulation components, composability emphasizes the need for a conceptual alignment of data among the components part of a simulation system. In this regard, the main objective of the proposed architecture is two-fold, namely the provision of an approach for the specification of the simulation system conceptual model, to identify the available existing M&S components, and the use of the TOSCA standard for describing both the M&S components and the composed simulation, in order to address interoperability issues.

In [25], it is argued that MSaaS-based applications should be easily composed by integrating loosely coupled shared components (in other words, simulation services), in a cloud-based environment. The pillars of the MSaaS ecosystem have been discussed in [26]. According to the proposed architecture, a MSaaS system should be constituted by *M&S services*, which are the building blocks of simulation applications, *registries*, and *repositories*, containing M&S services descriptions and implementations, respectively, *processes*, which define how services are discovered, composed, deployed and executed, an *infrastructure*, which describes the simulation environment and, finally, a *portal*, which constitutes the entry point to start the MSaaS process. Such contributions have inspired the essential building blocks at the basis of the ArTIC-MS's architectural design, which also introduces a step forward by addressing interoperability in the case of inter-cloud service composition.

Finally, existing literature also provides relevant examples of already available MSaaS platforms, such as CloudSME [27], Simulation Platform [28], and OCEAN [15]. Due to their architecture and rationale, CloudSME and the simulation platform are largely different from ArTIC-MS. CloudSME [27] is a multi-cloud platform for developing and executing commercial cloud-based simulations and its primary target audience includes commercial software vendors and consultant companies in the IT domain, as well as small and medium-sized enterprises (SMEs).

The simulation platform [28] consists of a cloud of virtual machines (VMs) upon which a GNU/Linux OS runs. Various types of software, including scientific software, compilers, libraries, and neural simulators are pre-installed on each VM. The platform allows users to request the assignment of a set of VMs to build and run a scientific simulation, according to specific requirements.

Differently, OCEAN [15] is a MSaaS platform based on OpenStack and specifically designed and developed for supporting training and simulation-based exercises in the context of NATO-funded military research programs. Similar to the ArTIC-MS platform, OCEAN's architecture includes a cloud infrastructure, a service repository, and a portal that allows users to discover, select, compose, and deploy M&S components. Moreover, it also shares the potential target audience. The main difference is that OCEAN is not specifically designed to support inter-cloud interoperability. Its orchestration engine (i.e., OpenStack HEAT [29]) requires the use of an implementation-specific technology, namely HOT (Heat Orchestration Template) [30], for the description of a simulation application, while ArTIC-MS makes use of the TOSCA standard.

Regarding the orchestration of service composition, the architecture proposed in [14] exploits MDE-based approaches and techniques to automate the application-level orchestration of M&S services according to an abstract composite service specified by use of different standards and notations, e.g., the unified modeling notation (UML) and the business process modeling and notation (BPMN). As already mentioned in Section 5, ArTIC-MS deals with the infrastructure-level orchestration, e.g., the execution of actions required for deploying and executing the several M&S services onto the required execution platform.

Finally, a preliminary version of ArTIC-MS was proposed in a previous work that this paper extends [12]. It should be underlined that the novel contribution this paper provides is not limited to a revision of ArTIC-MS building blocks. Differently, the two papers are quite different from each other and they do not address the same objectives. Moreover, the completeness of this work goes far beyond what we discussed in the previous contribution. Specifically, the previous paper aims to provide a preliminary description of ArTIC-MS, which focuses on the identification of architecture's building blocks. Such a description does not give any detailed view of the capabilities of each component and does not discuss the conceptual approach used for its specification. Moreover, the previous paper discusses an example application for assessing the feasibility and limitations of standards, principles, and technologies at the basis of the proposed approach.

Differently, this work illustrates the complete methodological and architectural approach that has been used to specify ArTIC-MS, as discussed in Sections 4.2 and 4.3, respectively. Moreover, this paper provides a complete description of ArTIC-MS from

functional and operational points of view, as outlined in Sections 5 and 6, respectively. Specifically, the architecture is described in terms of:

- The adopted conceptual architectural approach, clarifying how ArTIC-MS was derived from a reference architecture, and how concrete implementations might be based on ArTIC-MS;
- The capabilities of ArTIC-MS;
- The operational view of ArTIC-MS, which is described throughout the specification of ArTIC-MS actors and use cases that specifically address the federation of heterogeneous infrastructure.

From a concrete point of view, this work discusses an example application carried out as part of the MASTER research project, which addresses a federation of two heterogeneous cloud infrastructure (OpenStack [16] and Alien4Cloud [31]) to show how ArTIC-MS can be used for supporting the FAIR principles (findability, accessibility, interoperability, and reusability) of a MSaaS platform.

## 3. Background

This section briefly outlines the standards and technologies on the basis of the proposed contribution. Specifically, Section 3.1 illustrates the OASIS standard TOSCA, while Section 3.2 briefly introduces the open-source cloud implementation OpenStack.

### 3.1. Topology and Orchestration Specification for Cloud Applications (TOSCA)

The description of a cloud application's topology and its deployment and configuration is a complex task that cloud vendors address by adopting different approaches and technologies, such as the Amazon AWS CloudFormation [32] or the Heat Orchestration Template (HOT)) [30], the template format based on the YAML (YAML Ain't Markup Language) [33] markup language, which is supported by Heat [29] the orchestration engine of the OpenStack cloud platform [16].

In this context, in order to pursue the harmonization of existing approaches, the Organization for the Advancement of Structured Information Standards (OASIS) has proposed TOSCA (Topology and Orchestration Specification for Cloud Applications), a standard language for describing cloud-based service orchestration [13,34].

TOSCA defines a YAML-based specification for describing an IT service in terms of both its computing infrastructure and the required procedures for deploying, instantiating, executing and managing the service. The standard also specifies a packaged file format, namely Cloud Service Archive (CSAR), which allows one to store in the same package the YAML service description and the set of software artifacts (e.g., OS virtual images, libraries, scripts, DMBSs installation files, middleware, application software in executable form, etc.) that actually implement the service. The CSAR package is given as input to the *TOSCA Engine*, which is responsible for processing the YAML description so, overseeing the deployment of several artifacts composing the IT service, and orchestrating the execution of the related managing procedures.

In TOSCA a *service* is specified in terms of a *Service Template*, which is a YAML description, which completely specifies the service structure and its related characteristics. A service template includes the following elements:

- **Topology template**, which is the most relevant component of a service template, as it describes the structure of a service in terms of its building blocks. The service structure is specified by the use of a direct graph in which nodes represent the building blocks of a service (e.g., servers, network interfaces, virtual images, databases, etc.), and edges represent the relationships between nodes (e.g., deployment relationship, connection relationship, etc.). According to a hierarchical structure, in TOSCA nodes and relationships are, in turn, further specified by the use of *node templates* and *relationship templates*, respectively.
- **Node type and relationship type**, as in TOSCA each element must be associated with a type. Indeed, node templates and relationship templates are typed by node type

and relationship type, respectively, which provide the characterization of properties and interfaces of each TOSCA topological element. The Service template contains a separate description of node types and relationship types to facilitate their reuse.

- **Life cycle operations**, as TOSCA addresses the operational management of a service, the Service Template also includes the specification of executable artifacts (e.g., scripts) implementing the several operations executed by the TOSCA engine to handle the service during its life cycle (deployment, execution, 'undeployment', configuration, etc.).

As the use of TOSCA allows the specification of vendor-agnostic service orchestration, in this work, TOSCA is used as a reference notation for specifying reusable and interoperable service descriptions.

A detailed description of the TOSCA standard is beyond the scope of this paper. Interested readers are referred to the official documentation.

### 3.2. OpenStack

OpenStack [16] is an open-source project providing the implementation of an infrastructure-as-a-Service (IaaS) cloud infrastructure. OpenStack provides the orchestration engine HEAT [29] that allows the deployment and execution of composite cloud applications described by use of topology templates specified in the YAML-based description format HOT (Heat Orchestration Template) [30].

OpenStack also includes *HEAT Translator* [35,36], a component that allows the translation of TOSCA templates to semantically equivalent HOT templates. As clarified in Section 7, ArTIC-MS exploits the *HEAT Translator* to enact the interoperability of services provided by different vendors in a heterogeneous MSaaS federation. In this respect, it is worth noting that the HEAT Translator was designed to be easily extended to enable the translation of various input formats to HOT descriptions.

## 4. MSaaS Reference Architecture

This section illustrates the proposed reference architecture (RA) for MSaaS federated infrastructure. In order to better frame this paper's contribution and clarify the adopted terminology, Section 4.1 outlines the addressed context: the development of a distributed simulation as an orchestration of services in the cloud. Section 4.2 briefly outlines the proposed approach in the frame of the enterprise architecture design. Finally, Section 4.3 introduces the proposed MSaaS RA.

### 4.1. Orchestration of MSaaS-Based Distributed Simulations

According to a MSaaS perspective, a distributed simulation is an application built as an *orchestration* of existing M&S services available in the cloud. Specifically, the term *orchestration* takes into consideration both the specification of how services have to cooperate to provide the required functional behavior and the management of several components that contribute toward providing the execution platform. Thus, the development of a MSaaS application is typically undertaken through the following steps, as summarized in Figure 1:

- **Service discovery**: In order to satisfy the MSaaS application requirements, a service discovery is performed to identify the set of candidate M&S services providing the required capabilities. Specifically, as detailed in Section 5.2 the functional and non-functional services characteristics are described in terms of *metadata-based descriptions* stored in a services registry, while the related services implementations are stored in a services repository.
- **Service composition**: The service composition or, in other words, an *application orchestration*, refers to architectures, methods, and tools, used to coordinate the execution flow and the messages exchange among the M&S services identified in the previous step, in order to meet the functional requirements of the needed simulation.
- **Service deployment**: Each M&S service implementation requires to be deployed on top of the given execution platform, which consists of computing nodes, systems and

application software, databases, network connections, etc. The term *infrastructural orchestration* includes those activities needed to set up the execution infrastructure and deploy, configure and eventually start the required M&S services. In this context, machine-readable formats for specifying the service deployment description, along with an orchestration engine able to compute such descriptions, can be introduced for easing the service infrastructural orchestration.

With regard to infrastructure orchestration, this work exploits the TOSCA standard to improve the portability and interoperability of M&S services. This is achieved by specifying YAML-based and vendor-independent orchestration descriptions that can be automatically processed by any TOSCA-compliant engine.

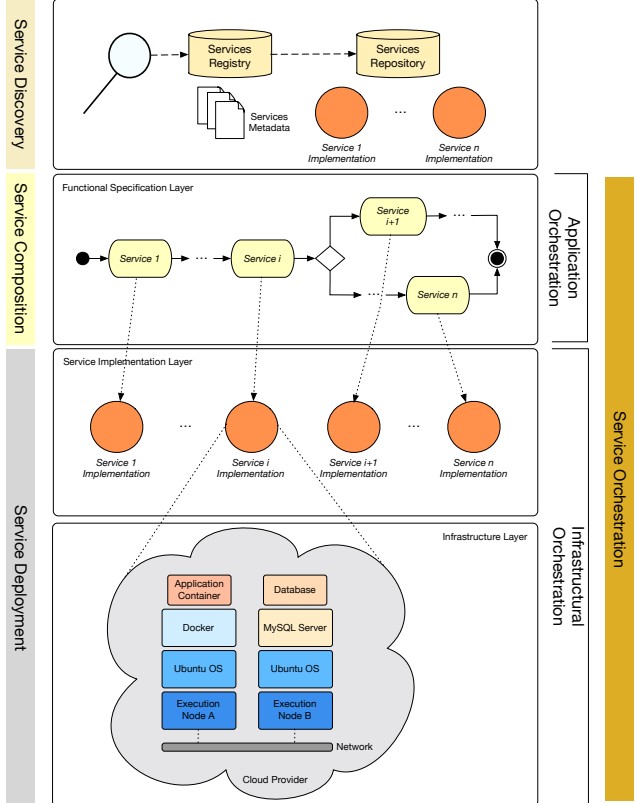

**Figure 1.** Application and Infrastructural orchestration for MSaaS applications.

### 4.2. Enterprise Architecture and Reference Architectures

The design and development of large and distributed systems built by integrating various components that cooperate by exchanging control information and data are non-trivial tasks. Thus, the specification of a *system architecture* (which specifies the system structure, the boundary of each component, the interfaces they provide, and the expected inputs and outputs) is one of the most important steps to be taken in developing a complex system.

More generally, in order to make the system architecture suitable to provide different stakeholders with the required information, various *architectural layers* have to be considered. *Conceptual architecture* models can be defined to describe the system from an abstract or business-related perspective. Then, iteratively, more refined architectural models are derived from abstract ones, until *concrete architectures* specifying the internal detail of each system component are finally developed. Several *Architectural framework* exist that provide guidance for structuring, classifying, and specifying such architectural layers, among others, NAF [37], DoDAF [38], and TOGAF [39].

In this context, the *enterprise architecture* discipline [40], introduces the concept of *reference architecture (RA)*: A RA is a part of the enterprise architecture that provides standards and documentation for a particular type of capability throughout the enterprise [41].

Thus, an RA is an *abstract architectural template* that provides guidance to support the development of a concrete system in a given operational context. More specifically, a RA addresses a given domain and describes an abstract architecture in terms of structural building blocks, their relationships, and the capabilities each building block provides.

From the same RA, one or more **concrete architectures** can be derived, each used for driving the implementation of actual systems, as summarized in Figure 2.

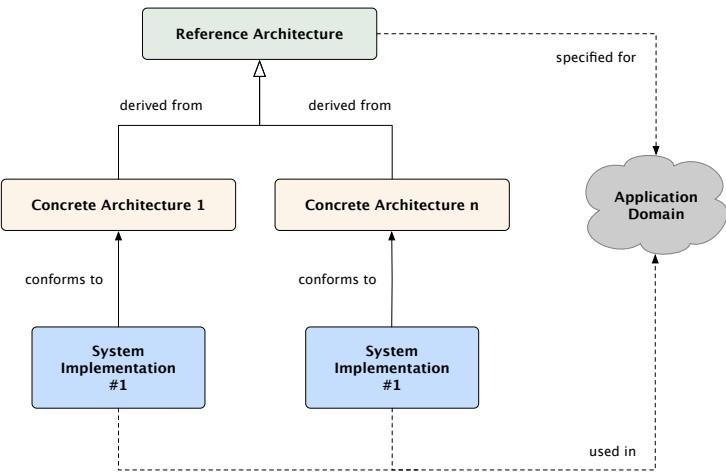

**Figure 2.** Reference architecture, concrete architectures, and systems implementations.

*4.3. Structure of the MSaaS Reference Architecture*

The proposed MSaaS RA, shown in Figure 3, consists of the following building blocks:

- **User interface:** Provides a web-based visual interface that allows users to make use of the capabilities provided by other RA building blocks.
- **Discovery service:** Provides the capability to perform a federated service discovery, in order to retrieve the existing M&S services made available by the federation's participant.
- **Composition Service:** Provides the capability to create a MSaaS application by composing the needed services.
- **Deployment service:** Provides the capability to carry out the following activities:
  - Configuration of each M&S service included in the composition;
  - Specification of the computing nodes constituting the execution platform;
  - Description of how the M&S services have to be deployed onto the execution platform.
- **Repository service:** Provides the capability for storing actual M&S services implementations.
- **Registry service: Provides the capability for storing M&S services descriptions.**
- **Orchestration service:** Provides the capability to run the executable artifacts included in the deployment description.
- **Adaptation service:** Provides the capability for adapting the deployment description, in order to make it compliant with the Orchestration Service.
- **Modeling and simulation services:** Provide the capability for building a MSaaS application. When a new M&S service is registered on the MSaaS infrastructure, the corresponding entry, which includes the service description, is created in the concrete registry managed by the registry service. Moreover, the service implementation is made persistent due to the capability provided by the repository service.
- **Simulation management and execution control (SMEC) Service:** Provides the capability for overseeing the simulation execution.
- **Infrastructure control service:** Provides a set of cross-functional capabilities including logging capability, infrastructure monitoring capability, resource monitoring capability, etc.

As discussed in Section 1, along with the MSaaS RA, this paper's contribution also includes ArTIC-MS, a conceptual architecture derived from the RA, which specifically introduces TOSCA as an enabling technology for ensuring the actual interoperability and portability of M&S services in a federated infrastructure.

The next section introduces ArTIC-MS and clarifies how the proposed approach ensures the interoperability of concrete architectures derived from ArTIC-MS or directly from the RA.

**Figure 3.** MSaaS reference architecture.

## 5. Architecture for the TOSCA-Based Inter-Cloud M&S (ArTIC-MS)

As stated in Section 2, this work extends a preliminary version of ArTIC-MS [12]. Specifically, Section 5.1 outlines the rationale at the basis of the ArTIC-MS design, Section 5.2 provides a detailed description of ArTIC-MS building blocks, Section 5.3 clarifies the role played by ArTIC-MS and TOSCA in a heterogeneous federation and, finally, Section 5.4 outlines how TOSCA features are exploited to build M&S services.

### 5.1. ArTIC-MS Rationale

In order to properly address the inter-cloud service orchestration scenario, the design of ArTIC-MS is based on principles and assumptions illustrated in [12] and hereby summarized for the sake of completeness:

- **Service descriptions**: The interoperable integration of services deployed in different infrastructures and implemented by the use of various technologies requires the availability of an agnostic and technology-independent service description. ArTIC-MS adopts service descriptions that specifically address:
  - *Metadata*: This is used for describing services. Due to the availability of metadata, users can identify the most suitable services that have to be integrated into the MSaaS application. Metadata are stored in the *service registry*.
  - *Infrastructural orchestration description*: This is used to deploy and execute a service. ArTIC-MS assumes that such a description, which is stored in a *Service repository*: This is provided as a TOSCA CSAR package. The scenario that clarifies how an ArTIC-based MSaaS implementation can be federated with other MSaaS infrastructure is discussed in Section 7.

- **Service discovery**: In federated MSaaS infrastructure, a discovery service shall be provided to enable users in identifying and retrieving services that each partner made available on its cloud platform. In this respect, it is assumed that the service registry provides an application program interface (API) for implementing inter-cloud service discovery.
- **Service availability**: The provisioning of various MSaaS services is the responsibility of the several federated infrastructure partners. In turn, each partner shall be provided with features to retrieve and integrate such services to build more complex MSaaS applications. In this respect, each service can be provided by a partner in two different configurations, according to the preferred business model:
  - *Running services*: A partner might provide a running service that is deployed and configured under the responsibility of the providing partner. In this case, the service metadata includes a service *endpoint* (e.g., a URI) and an interface description (e.g., by use of web service definition language (WSDL));
  - *deployable services*: A partner might provide an instance of the service specified in terms of artifacts and an infrastructural orchestration description (e.g., a TOSCA CSAR package). In this case, the service metadata includes a reference to the service repository that stores the CSAR package.
- **Service composability**: The main objective of the ArTIC-MS platform is to support users in executing the various activities required to build, describe and make available M&S services, and to use such services for building complex MSaaS application. It should be underlined how a *MSaaS application* can be treated as a (complex) M&S service; thus, its description and implementation will be stored in the service registry and the service repository, respectively. Moreover, it can be recursively used as a building block of other compositions.

*5.2. ArTIC-MS Conceptual Architecture*

ArTIC-MS conceptual architecture, which is shown in Figure 4, consists of the following building blocks:

- **Web-based user interface:** Provides a web-based visual interface that allows users to make use of the capabilities provided by other ArTIC-MS building blocks.
- **Composer:** Provides a visual environment to create the MSaaS application by composing the required services.
- **Deployment handler:** Provides the capability to carry out the following activities:
  - The configuration of required parameters for each concrete M&S service included in the MSaaS application;
  - The configuration of each computing node that composes the execution platform;
  - The specification of the required configurations specifying how M&S services have to be deployed onto the execution platform.
- **Repository interface:** Provides an interface to access the repository service.
- **Registry interface:** Provides an interface for accessing the registry service.
- **Federated service discovery:** Provides a visual environment that allows users to perform federated service discovery. This building provides the capability for executing inter-cloud M&S service discovery by exploiting the registry services API made available by each federation's participant.
- **Cloud infrastructure interface:** Provides the ability to interact with the underlying cloud infrastructure.
- **Services repository:** Provides the capability for storing actual M&S services implementations.
- **Services registry:** Provides the capability for storing M&S services descriptions.
- **TOSCA engine:** Provides the capability for supporting the MSaaS deployment and its execution. As mentioned in Section 3.1, the CSAR archive contains the YAML-based service template specification and the executable artifacts which actually implement

the service. In this respect, the TOSCA engine takes as input the CSAR description of each concrete service composing the MSaaS application and computes the related TOSCA service template in order to:

– Deploy the service implementation on top of the required execution platform, according to the topology template specification;
– Start the service by invoking the required life-cycle operations provided by the executable artifacts.

- **Service description translator:** Provides the capability for parsing and translating a YAML-based TOSCA service template to make it compliant with non-TOSCA cloud implementations, and vice versa.

To better clarify the relationship between the RA and ArTIC-MS it should be underlined that the RA specifies an abstract template for guiding the development of concrete MSaaS infrastructure implementations, while ArTIC-MS is a *conceptual architecture* which *refines* the RA in order to achieve the following objectives:

- The explicit adoption of TOSCA, which is acknowledged to be a promising standard for supporting the interoperability of M&S services in heterogeneous MSaaS federation;
- The specification of appropriate use cases, in order to describe how the composition, deployment and orchestration of M&S services in a federated MSaaS infrastructure is dealt;
- The identification of abstract components constituting the backbone of any MSaaS infrastructure implementation based on TOSCA.

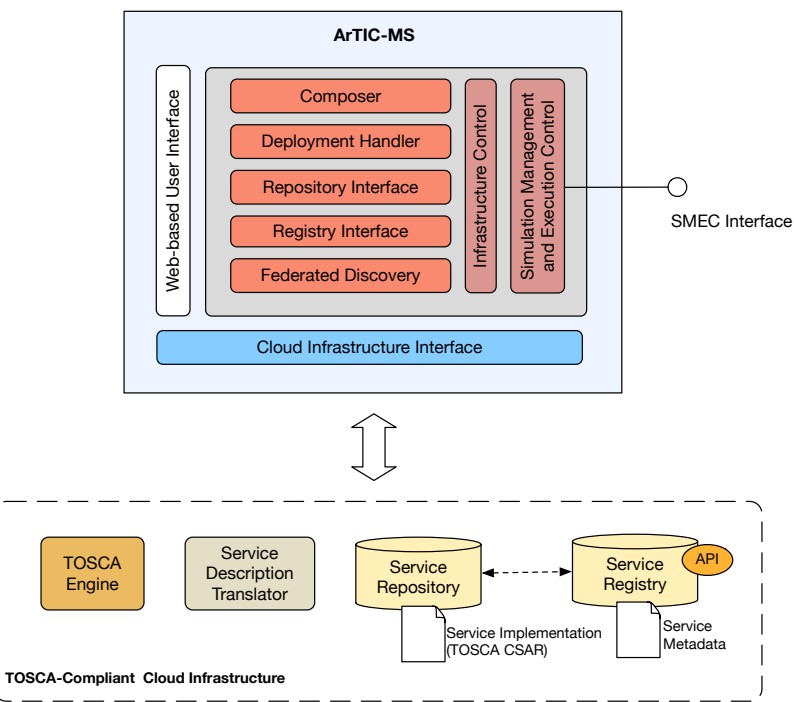

**Figure 4.** Conceptual architecture of ArTIC-MS.

Regarding the last point, it should be underlined that, in order to keep the proposed solution as abstract as possible, the RA itself has been specified in terms of interacting services that provide the required capability, according to the MSaaS paradigm. Differently, ArTIC-MS identifies appropriate *components* which have to be included in any concrete architecture deployed as part of the MSaaS infrastructure implementation. As an example, the RA introduces a *capability* provided by the *registry service* which, from a MSaaS perspective, might be available in a catalog of existing M&S services. Differently, ArTIC-MS includes a *registry component* that must be included in any compliant concrete architecture

and, consequently, implemented as an actual repository capable of storing YAML-based service templates.

Thus, the adoption of ArTIC refines the layered architecture principle introduced in Section 4.2, as shown in Figure 5.

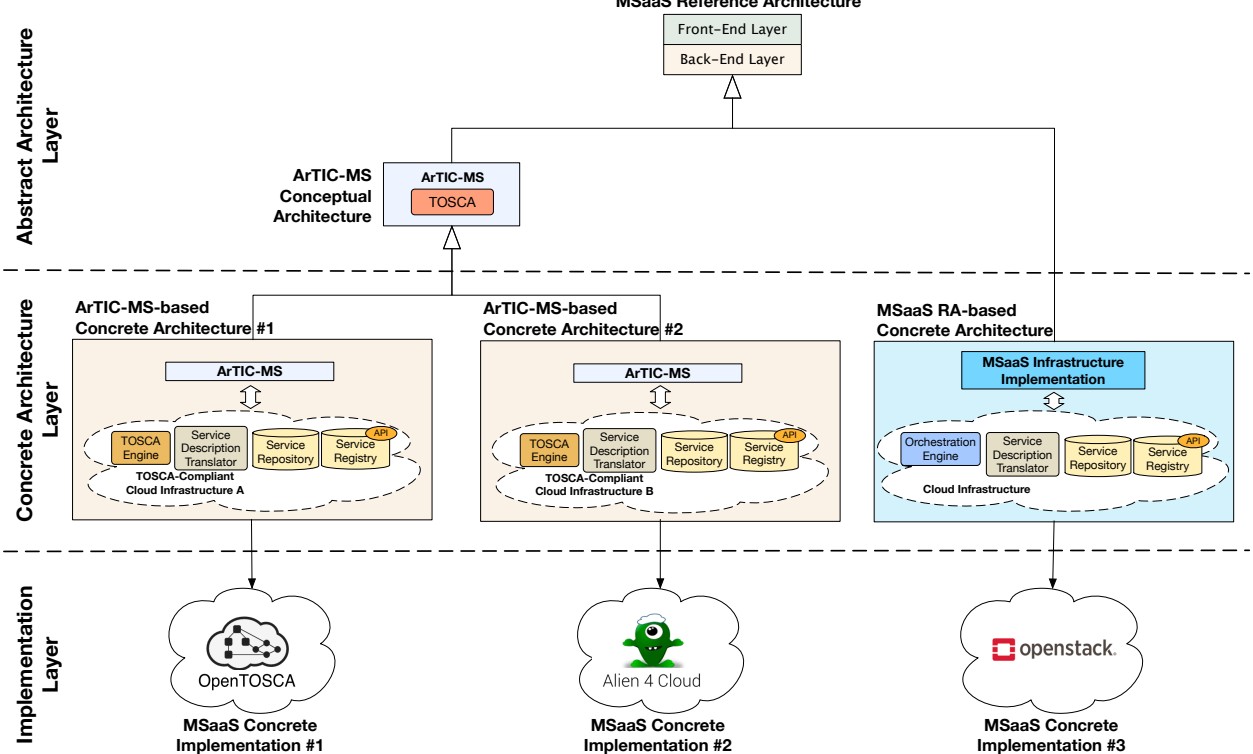

**Figure 5.** Relationships among different architectural layers.

Three different layers have to be considered:

- **Abstract architecture layer:** Describes an abstract architecture, which has been specifically defined to cope with a given application domain. In our approach this layer is further specialized, to include the two following architectural layers:
  - MSaaS reference architecture: As described in Section 4.3, the RA is the abstract template which any MSaaS concrete architecture has to comply with.
  - ArTIC-MS: A conceptual refinement of RA that introduces TOSCA.
- **Concrete architecture layer:** Includes any concrete architecture for implementing a specific MSaaS Infrastructure. TOSCA-oriented architecture shall be compliant with ArTIC-MS, while architectures based on different standards and technologies shall be directly derived from the RA.
- **Implementation layer:** Includes any concrete implementation of MSaaS infrastructure.

The next section further clarifies how different MSaaS infrastructure implementations might cooperate in a heterogeneous federation.

### 5.3. ArTIC-MS in a Federated MSaaS Infrastructure

The proposed architectural approach fosters the effective interoperability and portability of M&S services provided by different MSaaS infrastructures in a *heterogeneous* federation. Specifically, the adoption of TOSCA and compliance with the proposed MSaaS RA aims at effectively supporting such a scenario, as shown in Figure 6.

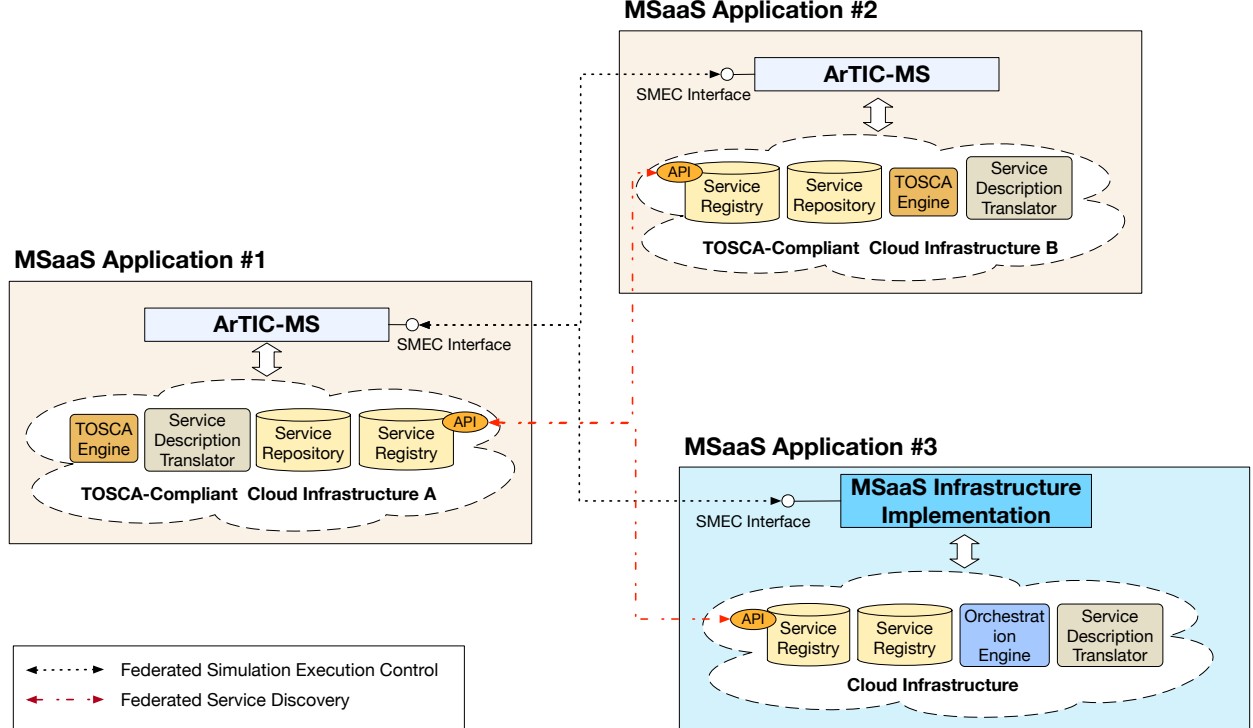

**Figure 6.** Federated MSaaS infrastructure.

The pillars upon which the federation is built are the following:

- A registry service providing an API-based discovery capability;
- The adoption of a machine-readable deployment description;
- The availability of a service description translator building block that provides the capability for translating deployment descriptions that are not specified by the use of TOSCA (and vice-versa);
- Orchestration engine which provides the capability for processing the adopted deployment description notation.

A detailed description that shows from an operational perspective how ArTIC-MS might support federated MSaaS infrastructure implementations is provided in Section 6.

### 5.4. Service Description and Composition in TOSCA

One of the founding pillars of ArTIC-MS is the availability of a catalog of M&S services that can be composed and orchestrated in order to build distributed MSaaS applications. In this respect, as the provisioning of M&S services constitutes the basis of the ArTIC-MS operational specification provided in Section 6, this section illustrates the TOSCA principles on which the classification and composition are based.

According to TOSCA grammar, a service template is a composable item: indeed, an existing `topology template` can be reused as the `node template`, which constitutes the building block of a larger and more complex service.

That principle is actually enacted by the *substitution mapping feature*. TOSCA allows the specification of *abstract* node templates, e.g., node templates that do not provide any implementations for the life-cycle management operations. At execution time, the TOSCA engine is provided with appropriate `substituting templates`. Such concrete templates provide the same external façade (i.e., properties, capabilities, etc.) as the abstract node template and also include a concrete specification for the remaining template elements, as shown in Figure 7.

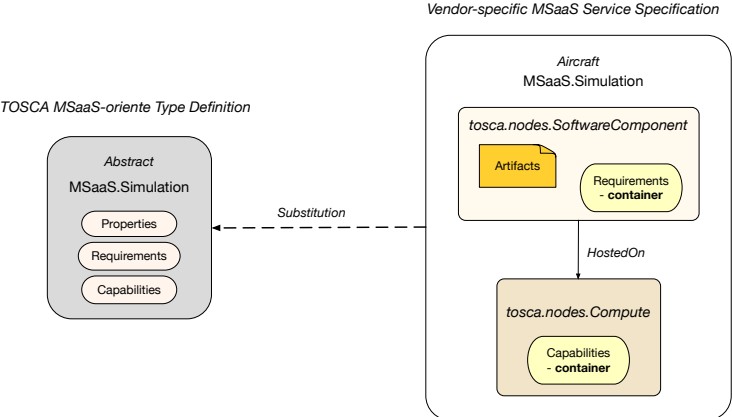

**Figure 7.** Service substitution principle.

The substitution principle is introduced in order to help defining a conceptual classification of available services and related capabilities. As explained in Section 6.1, ArTIC-MS specializes the services repository by introducing a MSaaS-oriented type repository, which stores abstract templates that any provider might use as building blocks for the concrete services they want to make available.

*5.5. Comparison with the Preliminary Version*

As stated in Section 2, this work extends the preliminary version of ArTIC-MS that has been proposed in [12]. In order to better point out the contribution proposed in this work, this section summarizes how ArTIC-MS has been revised. The preliminary version is depicted in Figure 8.

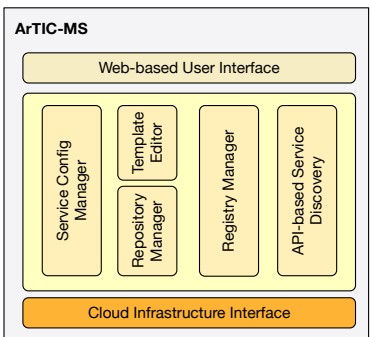

**Figure 8.** Preliminary version of ArTIC-MS.

The proposed revision of ArTIC-MS does not aim to disrupt what has been proposed in the previous work. Rather, in this work, the ArTIC-MS structure has been reorganized and rationalized in order to make it able to provide the capabilities required for its use in the addressed concrete experimentation. Moreover, since the previous work included only preliminary identification of ArTIC-MS components, without describing the relevant capabilities, this work provides a completely functional and operational specification of the architecture. Specifically, compared to [12], this paper's Section 5:

- Outlines the rationale of the architecture;
- Provides an extensive description of the capabilities that each component shall provide;
- Introduces new components (composer, deployment manager, federated service discovery), which replace existing ones, revising and enriching the provided capabilities;
- Introduces new components (service description translator, infrastructure control and simulation management and execution control), which have not been considered in the preliminary version;

- Revises the description and the provided capabilities of the repository manager and the registry manager.

## 6. ArTIC-MS Use Case Specification

This section describes ArTIC-MS from an operational perspective. In this respect, the UML use case diagram shown in Figure 9 outlines the responsibilities of the following ArTIC-MS users.

- **Simulation expert (SE):** Responsible for the elicitation and specification of MSaaS application requirements.
- **Simulation service provider (SSP):** Responsible for the development and the provisioning of M&S services;
- **Simulation developer (SD):** Responsible for the development of MSaaS applications. Specifically, the SD deals with the identification of appropriate M&S services and their composition to build a MSaaS application compliant with the simulation requirements provided by the SE;
- **TOSCA expert (TE):** Owns appropriate knowledge of the TOSCA standard. She/he is also in charge of supporting other users for the specification of the required YAML-based templates;
- **Simulation user (SU):** Final user of the simulation application;
- **ArTIC-MS administrator (ADM):** The user who possesses the required skills for managing the ArTIC-MS platform. Its main task is to provide other users with the required environment for building, executing, and monitoring simulation experiments.

The following sections detail the use cases that focus on the development and deployment of a MSaaS simulation built as an orchestration of available M&S services. It is worth noting that this paper only considers the case in which the federated MSaaS infrastructure includes heterogeneous cloud implementations, as discussed in Section 5.3. A scenario dealing with all TOSCA-based cloud implementations has already been discussed in previous work [12]. Finally, the use cases dealing with ArTIC-MS management and simulation experiment execution/evaluation are not further discussed in this work.

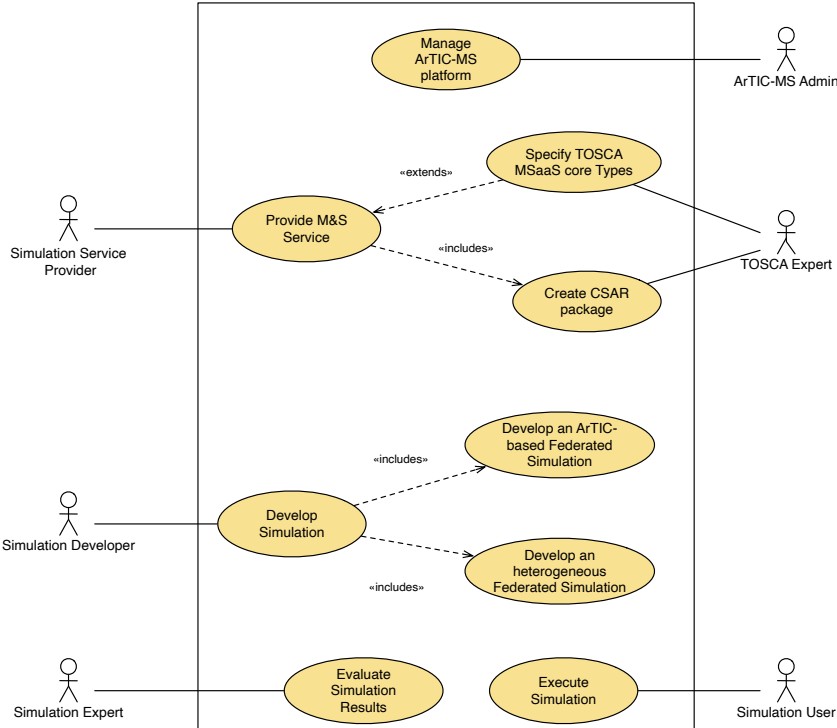

**Figure 9.** ArTIC-MS use case diagram.

*6.1. Development and Provisioning of a M&S Service*

This use case, whose execution flow is outlined in Figure 10, involves two users owning different skills: the TOSCA expert (TE) and the simulation service provider (SSP).

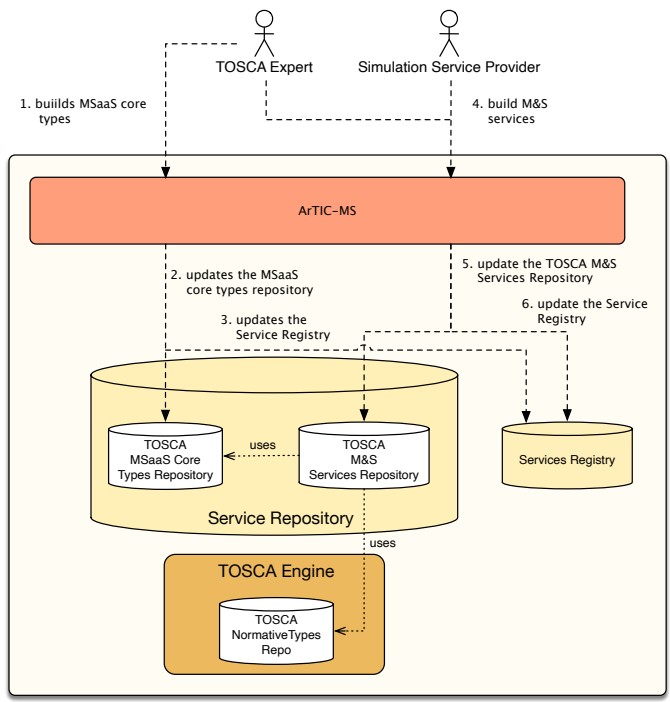

**Figure 10.** Use case for building a TOSCA-based M&S Service.

As discussed in Section 5.4, a *service template* is composed of *typed* elements (e.g., nodes and relationships) and might include other existing (and abstract) templates. Thus, the SSP makes use of existing types and templates that are refined, extended, and composed in order to develop the needed M&S service. Differently, the TE is responsible for the specification of the *TOSCA core MSaaS templates* and also supports the SPP to specify the various YAML-based TOSCA templates needed to make available the provided M&S services.

In this respect, Figure 10 also provides a more detailed view of the following ArTIC-MS building blocks:

- **TOSCA engine:** Provides the implementation of *TOSCA normative types* which constitutes the building blocks of other templates;
- **Services repository:** Includes the *TOSCA MSaaS core types repository*, which stores the abstract and concrete templates for the *core* MSaaS types upon which other service templates are built, and the *TOSCA M&S services repository*, which hosts the CSAR packages wrapping the provided *M&S services*, respectively.

The use case is specified as follows:

1. The TOSCA Expert (TE) logs into ArTIC-MS to create the required templates for specifying *MSaaS core types* used to build the M&S services;
2. The YAML templates are stored in the *TOSCA MSaaS core template repository* and corresponding entries are added to the services registry;
3. The simulation service provider (SSP) develops the executable artifacts implementing the provided M&S services;
4. The TE and SSP cooperate to specify an appropriate YAML template for each provided M&S services;
5. The related CSAR package is built and stored in the *TOSCA M&S service repository*;
6. The SSP provides a metadata description for each M&S service that is stored in the services registry.

### 6.2. Composition and Deployment of a Federated Simulation Application

This section illustrates the use case in which the federation is composed by heterogeneous MSaaS infrastructure based on the RA. Specifically, the *service provider* is responsible for creating M&S services (and also MSaaS applications exported as CSAR complex services, as discussed in Section 6) by using the ArTIC-MS platform (left part of the figure). Differently, *Simulation Developers*, which uses a non-TOSCA MSaaS implementation, needs to build a MSaaS simulation by orchestrating local M&S services with TOSCA-based M&S services provided by the service provider. The scenario is specified as follows:

**Precondition**: *the Service Provider (SP) created a set of M&S services and MSaaS applications, which are available as CSAR packages in ArTIC-MS (see Figure 11—steps 1 to 4).*

1.  the *Simulation developer (SD)* logs into the MSaaS platform and executes a query to identify the available M&S services. As the query makes use of the API interfaces provided by each partner's registry, the *SD* is able to discover local services along with the TOSCA-based services provided by the *SP* (see Figure 11, steps 5);
2.  SD identifies the set of needed services (see Figure 11, step 6). In this respect, SD identifies $N$ services, where $N_L$ are available in the local infrastructure, $N_E$ services are provided by remote partners via a URI endpoint and an interface specification (e.g., WSDL, etc.); finally, $N_R$s are available from SP's remote services repository as CSAR packages, being $N = N_L + N_E + N_R$. In this respect, according to the business model assumed to be adopted by the SP, a subset of the $N_R$ (i.e., $N_{R-REM}$) services might need to be deployed onto the SP infrastructure, while the remaining $N_{R-LOC}$ can be retrieved and deployed onto the SI infrastructure, being $N_R = N_{R-REM} + N_{R-LOC}$
3.  SD retrieves from the local repository the locally available $N_L$ services;
4.  In order to ask for the deployment and execution of the $N_{R-REM}$ services that need to be directly managed by the SP's infrastructure, SD forwards a request to the ArTIC-MS *simulation management and execution control* service (see Figure 11, step 7);
5.  SD retrieves the CSAR descriptions of $N_{R-LOC}$ M&S services from the ArTIC-MS service repository (see Figure 11, step 8);
6.  SD translates the TOSCA template contained in the CSAR package to a vendor-specific template via the *Adaptation Service* (see Figure 11, step 9);
7.  SD uses the *composition service* to create a template that describes the given MSaaS application, by composing the retrieved services (see Figure 11, step 10);
8.  SD configures the required parameters of the various M&S services;
9.  SD configures the composition to make it possible to invoke the operations provided by the endpoints of the remote M&S services;
10. SD develops and configures a *deployment description* suitable for the underlying orchestration technology;
11. the *deployment description* is given as input to the *deployment service* in charge of executing the automated deployment of the simulation (see Figure 11, step 11);
12. The MSaaS application is finally ready to be used by the simulation end user (SU).
13. During the simulation execution, the appropriate interaction between the two cooperating MSaaS infrastructure is managed through the simulation management and execution control (SMEC) service (see Figure 11, step 12);

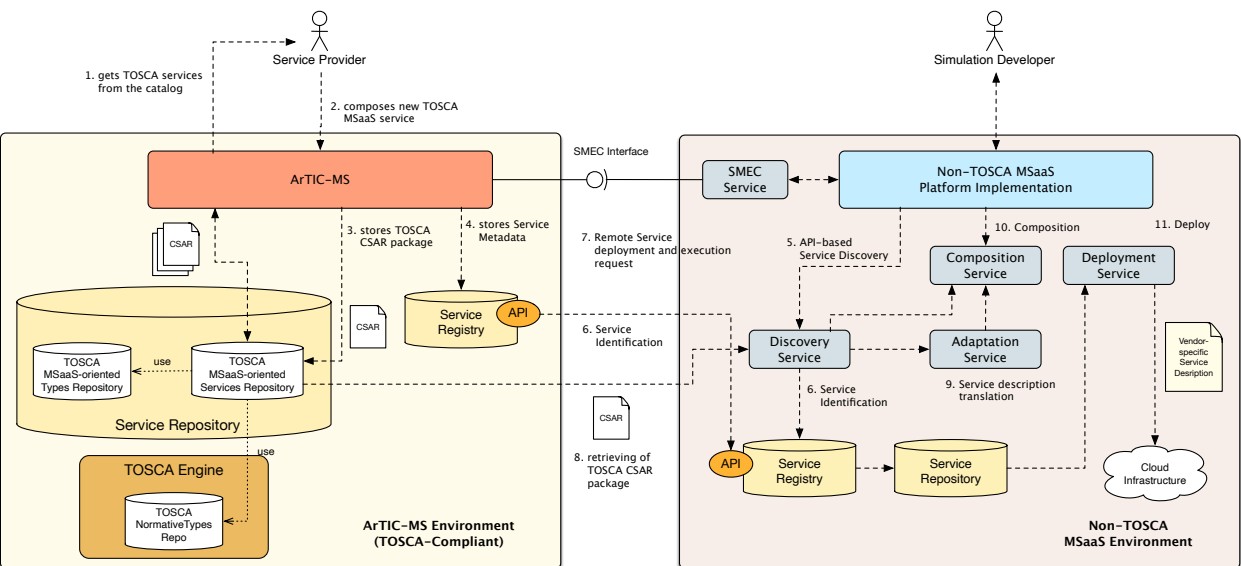

**Figure 11.** Federation of heterogeneous MSaaS infrastructure.

## 7. Experimentation of ArTIC-MS: Federation with OCEAN

As discussed in Section 1, the main objective of this paper is the complete specification of ArTIC-MS, a conceptual architecture aiming to foster the FAIR principles in MSaaS environments.

In this respect, a preliminary evaluation of the soundness of principles on the basis of ArTIC-MS has been discussed in [12]. Specifically, such an analysis has addressed the following issues:

- The availability of existing software products for developing TOSCA templates;
- The availability and usability of IaaS cloud implementations compliant with TOSCA;
- The effectiveness of the HOT translator component, for mapping TOSCA templates to HEAT-based ones.

Differently, in this work, experimentation was carried out as part of MASTER, a research project developed under the National Program for Defense Research, which aims to define innovative solutions for developing M&S applications in the cloud.

A prototype MSaaS concrete implementation based on the TOSCA-compliant IaaS platform Alien4Cloud [31] and compliant with ArTIC-MS has been developed. Then, a heterogeneous MSaaS federation has been built by federating the ArTIC-MS prototype with OCEAN [15], a MSaaS platform based on the open-source OpenStack cloud infrastructure [16].

As ArTIC-MS focuses on infrastructure orchestration, the experimentation has not addressed the evaluation of measurable simulation-related performance parameters (e.g., the simulation execution time or the speed-up). Rather, it has been conducted to assess how ArTIC-MS is able to support the FAIR principles in the development of a MSaaS-based application, by (i) easing the discovery of services in a heterogeneous federation (findability), (ii) retrieving the required M&S resources throughout a metadata-based discovery (accessibility, and (iii) improving the interoperability and reusability of M&S resources thanks to the adoption of TOSCA.

The experimentation has addressed the simulation of a maritime defense scenario, as depicted in Figure 12. The simulation includes two M&S services: a *scenario generator* available from the TOSCA-based ArTIC-MS prototype implementation, and a *naval simulator*, which is managed by OCEAN.

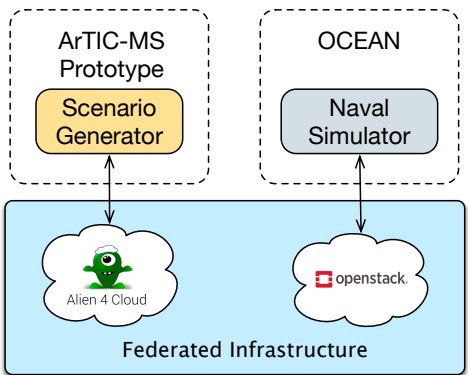

**Figure 12.** Federated MSaaS Infrastructure for the simulation of a maritime defense scenario.

The architecture of the adopted OCEAN implementation is depicted in Figure 13.

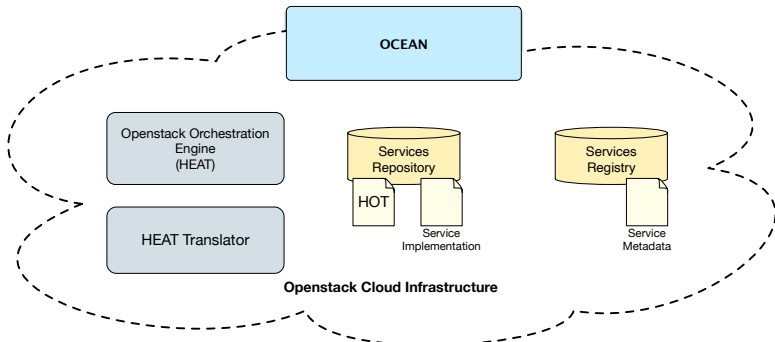

**Figure 13.** OpenStack-based OCEAN implementation.

OCEAN provides a visual interface to specify templates by composing the several elements (software artifacts, computing nodes, network connections, etc.) available from a dedicated toolbox and includes a feature for searching existing services that can be used as building blocks of more complex composed services. As an OpenStack-based implementation, OCEAN makes use of the orchestration engine HEAT to parse topology templates specified by the use of the HOT description format, which is based on YAML.

The experimentation addresses the use case discussed in Section 6.2 and specifically focuses on the portability of TOSCA-based M&S services onto the OCEAN infrastructure. Specifically, Figure 14 shows the sequence of activities and the flow of information exchanged between the two federated infrastructures.

*User A*, which acts as a *service provider*, is responsible for developing the *scenario generator* service, specifying the relevant TOSCA-based template and creating the CSAR package to make the M&S available to other users, as discussed in Section 6, by using the ArTIC-based infrastructure platform (upper part of the figure).

*User B*, which acts as the simulation developer, is responsible for developing the required simulation which addresses the maritime defense scenario illustrated in Figure 12, by identifying, retrieving, deploying, and orchestrating the required M&S services, e.g., the *scenario generator* and the *naval simulator*.

Specifically, the experimentation has been carried out as follows:

***Precondition:*** *executable artifacts implementing the Scenario Generator service are available for User A.*

1.  User A creates the CSAR package, which wraps the *scenario generator* service template and the relevant executable artifacts, and uploads it to the service repository. The relevant metadata are also uploaded to the service registry (see Figure 14 steps 1–3);
2.  User B logs into the OCEAN platform and executes a keywords-based query to discover the M&S services required for simulating the addressed scenario. Specifically,

the service registry is inquired to identify the services whose metadata satisfy the search criteria specified by User B (see Figure 14 step 4). The *naval simulator* service is available in the local infrastructure. Thanks to the capabilities provided by the federated service discovery component of ArTIC-MS, User B is able to discover the *scenario generator* hosted by the federated infrastructure;

3. User B retrieves the CSAR descriptions of the *scenario generator* M&S service provided by User A and stored in the ArTIC-MS repository (see Figure 14 step 5);

4. The TOSCA template contained in the CSAR package is translated to a HOT template (see Figure 14 step 6);

5. The generated HOT template is configured and deployed to the OCEAN infrastructure.

6. User B uses the visual environment provided by OCEAN to specify and configure the orchestration and, finally, executes the simulation (see Figure 14 steps 7–8).

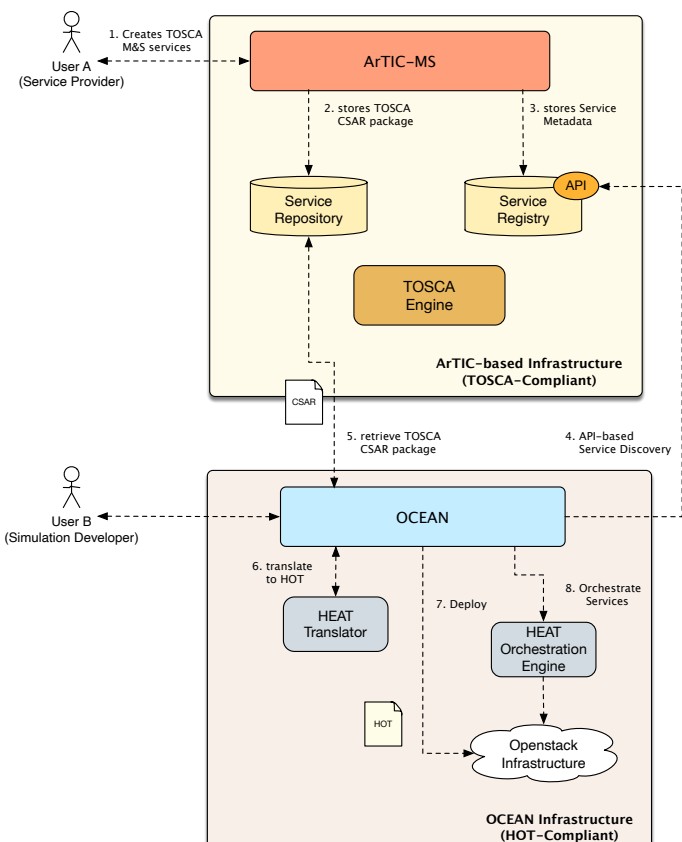

**Figure 14.** Federation of ArTIC-MS and OCEAN.

The *scenario generator* service provided by User A is available as a .iso virtual image to be deployed on top of a virtual machine. The server configuration includes two different network interfaces. A fragment of the TOSCA-based YAML service template is provided in Listing 1, while a fragment of the HOT-based YAML template yielded as output by the HEAT translator is provided in Listing 2.

**Listing 1.** TOSCA Service Template.

```
tosca_definitions_version: tosca_simple_yaml_1_0
description: scenario generator
topology_template:
...
node_templates:
virtual_machine:
type: SoftwareComponent
artifacts:
vm_image:
file: images/service-image.iso
type: tosca.artifacts.Deployment.Image.VM.iso
requirements:
host: server
# VM deployment
interfaces:
Standard:
create: vm_image
server:
type: tsca.nodes.Compute
capabilities:
host:
properties:
disk_size: 15 GB
num_cpus: 1
mem_size: 4096 MB
...
```

**Listing 2.** HOT Service Template.

```
heat_template_version: 2013-05-23

description: >
TOSCA_Scenario_Generator
...
virtual_machine_create_deploy:
type: OS::Heat::SoftwareDeployment
properties:
config:
get_resource: virtual_machine_create_config
server:
get_resource: server
flavor: m1.medium
user_data_format: SOFTWARE_CONFIG
networks:
- port: { get_resource: port1 }
- port: { get_resource: port2 }
virtual_machine_create_config:
type: OS::Heat::SoftwareConfig
properties:
group: script
config:
get_file: vm_image
...
outputs: {}
```

The next section discusses the experiment's results.

*Results and Discussion*

The experimentation has demonstrated the feasibility of the proposed approach and also showed how ArTIC-MS might support the development of a MSaaS-based distributed simulation in a federation of heterogeneous cloud infrastructure.

The capabilities provided by the proposed conceptual architecture allow the simulation developer to discover and retrieve remote M&S services available in federated infrastructure.

The adoption of TOSCA for describing services in the cloud in a vendor- and technology-independent fashion constitutes a relevant opportunity for supporting any MSaaS efforts. TOSCA might constitute the backbone of MSaaS implementations that increase the portability and interoperability degrees of M&S services in federated heterogeneous infrastructure, also fostering service reuse in different application scenarios.

On the other hand, the experimentation, along with the preliminary evaluation conducted in [12], has also revealed some limitations, which are mainly due to the maturity level of the TOSCA-related technologies and the limited support of the TOSCA standard.

It must be pointed out that TOSCA is not currently supported by most of the relevant players in the cloud marketplace (e.g., Amazon, Google, and Microsoft). Moreover, in a few years the underlying standard adopted for specifying TOSCA templates has evolved from XML (extensible markup language) to YAML and, besides that, the TOSCA conceptual model has also been slightly revised. While this fact demonstrates the vitality of the TOSCA developers community, the lack of a road map that clearly outlines how the standard will be evolved in the next future might constitute limitations in its widespread adoption.

Regarding the supporting tool for converting TOSCA templates to templates based on different languages (and vice versa), the availability of the HEAT Translator is a relevant starting point that we exploited for pushing the reusability and the portability of M&S services. Nonetheless, several efforts have to be spent in this direction. Currently, the HEAT Translator only supports the translation from TOSCA to HOT, while the opposite translation has to be addressed by ad-hoc extensions. Moreover, regarding the usability of HEAT Translator, in case of translation errors, the tool lacks verbose messages, which might help users to investigate and identify the specific problem.

## 8. Conclusions

This work investigated architectural approaches to support the FAIR principles in the development of heterogeneous federations of MSaaS infrastructure. The development of distributed simulations built as orchestrations of M&S services is a complex task that becomes even more difficult when the service composition includes M&S services provided by the heterogeneous infrastructure. In order to easily address this objective, the identification of an appropriate architecture assumes paramount relevance.

In this context, this work first proposes a reference architecture for identifying building blocks and relevant capabilities that a concrete MSaaS infrastructure architecture should provide to effectively deal with MSaaS federations. Moreover, this work investigated the role of TOSCA as a valuable standard to foster M&S service interoperability and portability in heterogeneous federations. In this respect, ArTIC-MS, a conceptual architecture based on the proposed RA, was introduced.

In order to provide effective guidance for implementing concrete MSaaS infrastructure in a federated environment, ArTIC-MS exploits the TOSCA standard and includes the specification of actors and use cases to define the relevant operational view.

This work also discussed experimentation focused on a maritime defense scenario built by orchestrating services in a heterogeneous MSaaS federated infrastructure.

According to the experimentation results, the capabilities provided by ArTIC-MS support the findability, accessibility, interoperability, and reusability of M&S services in heterogeneous infrastructure. In this respect, TOSCA constitutes a promising standard for dealing with service portability and their interoperable orchestration in a federated MSaaS infrastructure. Relevant issues that require further investigations include the actual adoption of TOSCA in significant and large-sized implementations, in both industry and academia, as well as the availability of open-source and commercial tools for translating various languages for services deployment and orchestration in the cloud.

Further work includes the development of a complete ArTIC-MS implementation based on a TOSCA-compliant cloud infrastructure and its evaluation in a heterogeneous federation. Moreover, additional effort is required to investigate methods, tools, and

languages for *application orchestration*, in order to appropriately extend the RA *composition service* and the ArTIC-MS *composer* building blocks.

**Author Contributions:** Conceptualization, P.B. and A.D.; investigation, P.B. and A.D.; methodology, P.B. and A.D.; resources, P.B. and A.D.; supervision, A.D.; validation, P.B. and A.D.; writing—original draft, P.B. and A.D.; writing—review and editing, P.B. and A.D. All authors have read and agreed to the published version of the manuscript.

**Funding:** This research received no external funding.

**Institutional Review Board Statement:** Not applicable.

**Informed Consent Statement:** Not applicable.

**Data Availability Statement:** The study did not employ or report any data.

**Conflicts of Interest:** The authors declare no conflict of interest.

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
