# Peer review of "A TOSCA-Based Conceptual Architecture to Support the Federation of Heterogeneous MSaaS Infrastructures†"

_futureinternet, doi:10.3390/fi15020048_

Round 1

Reviewer 1 Report

This paper is quite well written but there are issues which must be addressed:

1) What are the contributions of this work? In the introduiction section, It is absolutely unclear what are the new ideas that this work brings. The authors should provide their contributions by the end of the introduction section.

2) I am really concerned about the Experimental Results Section. I can't see how the proposed architecture improves the preliminary version of the conceptual architecture which was proposed in [7].  Also, i do not see why the authors claim that "The experimentation has demonstrated that the proposed architectural approach is feasible and promising. In what sense?

The whole Experimental Results section needs to be rewriten. The reader needs to see experiments conducted by the initial (or other approaches) and then see the comparisons to the new approach. Otherwise, it is not easy to verify the feasibility or any improvements.  Also, these experiments should highlight the contributions of this paper (which must be added).

Reviewer 2 Report

The article is generally well written and approaches interesting technical topics from the area of MSaaS infrastructures.

The paper can be seen as holding good enough value and probably may be even accepted for publishing in its current form. However, there are several issues worth mentioning with the aim to bring some overall improvements.

Let’s start with a fundamental one. From my perspective, there is a main concern regarding the current paper. This concern is coming from the fact that right before the Abstract the authors acknowledge that this is “an extended version of our paper published in Winter Simulation Conference 2020”. Future Internet Journal specifies the following:

“Your manuscript should not contain any information that has already been published.”

“(4) authors are asked to disclose that it is conference paper in their cover letter and include a statement on what has been changed compared to the original conference paper.”

For details, see: https://www.mdpi.com/journal/futureinternet/instructions

Thus, it becomes mandatory that the novelty of this paper must be much thoroughly explained throughout paragraph/section 2. It is true that the authors made an attempt on doing that, but still it remains rather unclear what are the new contributions in comparison with the previously published material? I suggest that the authors should provide the full text of their initial paper to be analyzed by reviewers and editor in order to be able to properly comprehend the added level of novelty and to dissipate any shadow of doubt in this regard. From the References section I could only have access to the Abstract of the previously mentioned published paper, not to its full content.

Now let’s move on and look at some other issues, in their order of appearance in the paper:

Line 9 – In the Abstract section the text states: “…in an federated environment” (the correct form is “a federated”)

Line 148 – please briefly explain the acronyms UML and BPMN as perhaps not all readers may know their meaning. Also, in the text BPMN is referred as being a “notation”, when a better term for this would be “standard”, as BPMN is widely accepted as the standard in Business Process Modeling.

Line 180 – please explain the acronym YAML

Line 243-244, on Figure 1 – please mention the origin of the presented model (whether it’s entirely authors’ contributions or not). The same for Figure 2 and Figure 3.

Line 313 - On paragraph / section 5 the authors talk about ArTIC-MS architecture. Please state here very clear which was the preliminary version from the previously published article so the differences can be unmistakably identified. Please include a figure with the former model and make an A / B comparison between each component so to ensure that the current work has brought a significant level of originality, novelty and value and may be accepted for publishing.

Line 468 – please correct the error from “…they want to made available”

Line 528 – Figure 10, and Line 568 in the text, please explain the SMEC acronym. The same for API acronym and CSAR. Please check carefully if there are also other unexplained acronyms used throughout the paper (e.g., OCEAN – first mentioned on Line 67 etc.).

Line 624 - Paragraph 7.1 – Results and Discussion, the text states: “The experimentation has demonstrated that the proposed architectural approach is feasible and promising”. This so-called "demonstration" is somehow still cluttered. More emphasis on this part is needed. More concise, what are the benefits? What will be the practical and maybe even managerial implications of the proposed (and improved) version of the conceptual architecture? So, more details in this regard would be beneficial for the overall quality of the paper.

Line 634 – the same, please explain the acronym XML. As it happens that I am familiar with all these acronyms, there is a chance that other readers may not be.

Again, please see the link: https://www.mdpi.com/journal/futureinternet/instructions where it states clear:

“Acronyms/Abbreviations/Initialisms should be defined the first time they appear in each of three sections: the abstract; the main text; the first figure or table. When defined for the first time, the acronym/abbreviation/initialism should be added in parentheses after the written-out form.”

Reviewer 3 Report

This paper is about how to build Modeling & Simulation (M&S) applications by integrating and orchestrating existing M&S services, with the objective of enhancing interoperability, composability, reusability and cost-effectiveness. The authors proposed an M&S-as-a-Service (MSaaS) Reference Architecture (RA) to ease the building of inter-cloud MSaaS applications. An extension of RA is also presented. It is called ArTIC-MS, which is a conceptual architecture that refines the proposed RA for introducing the TOSCA (Topology and Orchestration Specification for Cloud Applications). The work is interesting. But the presentation must be further improved.

1) The authors wrote: “Modeling & Simulation (M&S) is a widely used approach for analyzing systems, natural phenomenons and processes … .” But how is M&S related to Digital Twins and cyber-physical systems, which are related but more recent buzzwords? At least some discussion is in order.

2) The proposed architecture is based on Model-Driven Engineering (MDE)  to automate the application-level orchestration of M&S services according to an abstract composite service specified by Domain-Specific Languages (DSLs). However, MDE and DSLs are not well introduced and explained in the paper (DSLs are not even mentioned in the paper). The references on MDE and DSLs are almost non-existent. The authors should explain how MDE and DSLs are used in the proposed work.

3) What kind of analyses are performed from YAML specifications (e.g. Listing 2)?

4) There is a lack of technical details on implementing the proposed approach. More information on how MDE and DSLs have been applied is needed.

5) Figures 5-6 are too small, and the text is not readable.

6) Inconsistent notation: M§Services vs M&S Services.

7) Incomplete sentence on page 16: “… as outlined in the conceptual model provided in”.

8) Typos:

phenomenons

->

phenomena

run to reproduce the behaviour of the system o process under study

->

run to reproduce the behaviour of the system or process under study

// missing reference twice on page 4

HOT (HEAT Orchestration Template) [? ],

deploued

->

deployed

References used in this review:

===============================

Dalibor et al. 2022: Generating customized low-code development platforms for digital twins. Journal of Computer Languages, Volume 70, June 2022, 101117

Bano et al. 2022: Process-aware digital twin cockpit synthesis from event logs. Journal of Computer Languages, Volume 70, June 2022, 101121

Mohamed et al. 2020: Applications of model-driven engineering in cyber-physical systems: A systematic mapping study. Journal of Computer Languages, Volume 59, August 2020, 100972

Vještica 2021: Multi-level production process modeling language. Journal of Computer Languages, Volume 66, October 2021, 101053

Lelandais et al. 2019: Applying model-driven engineering to high-performance computing: Experience report, lessons learned, and remaining challenges. Journal of Computer Languages, Volume 55, December 2019, 100919

Ramos et al. 2022: Hapi: A domain-specific language for the declaration of access policies. Journal of Computer Languages, Volume 72, October 2022, 101153

de la Vega et al. 2020: Lavoisier: A DSL for increasing the level of abstraction of data selection and formatting in data mining. Journal of Computer Languages, Volume 60, October 2020, 100987

Round 2

Reviewer 2 Report

The paper has been improved and most of the issues have been addressed.

Reviewer 3 Report

The authors satisfactorily addressed my comments, and the paper can be accepted now. During the proofreading, the following statement in Section 3 should be rephrased: “As already mentioned in Section 5, … .” At this point, a reader did not read Section 5 yet.